∂ | Open Peer Review | Biotechnology | Research Article

# Study on *Lacticaseibacillus casei* TCS fermentation kinetic models and high-density culture strategy

Chen Chen,[1] Tianyu Guo,[1] Di Wu,[1] Jingyan Shu,[1] Ningwei Huang,[1] Huaixiang Tian,[1] Haiyan Yu,[1] Chang Ge[1]

**ABSTRACT** This study enhanced the production efficiency of Lacticaseibacillus casei TCS by optimizing medium composition and fermentation conditions for high-density culture. Initially, single-factor and orthogonal experimental designs and the response surface methodology were used to determine the optimal concentrations of medium. Subsequently, we applied the artificial neural network–genetic algorithm optimization method, which significantly increased the viable bacterial count. Fermentation kinetics were modeled using logistic growth and Luedeking–Piret models, which accurately predicted cell growth. Amberlite IRA 67, an anion exchange resin, effectively adsorbed lactic acid and maintained pH levels. Furthermore, the combined use of fed-batch fermentation and ion exchange alleviated the effects of acid inhibition, salt stress, and substrate limitation, resulting in a maximum cell density of 10.01 lg CFU/mL, a 9.3-fold increase over the basal medium. This study develops a robust and cost-effective strategy for the industrial production of *L. casei* TCS, significantly optimizing probiotic production processes.

**IMPORTANCE** *Lacticaseibacillus casei* TCS possesses outstanding aromatic characteristics, making it suitable for producing fermented dairy products. The goal of cultivating *Lacticaseibacillus casei* TCS at high density is to increase production yields, overcome challenges related to acid inhibition, and optimize fermentation processes. This study employed an artificial neural network (ANN) and genetic algorithms (GA) to determine the ideal composition of the proliferation medium for *Lactobacillus casei* TCS. It constructed a dynamic model to track bacterial growth, product formation, and substrate consumption during fermentation, analyzing the process's dynamic patterns. Furthermore, by utilizing resin adsorption and fed-batch cultivation techniques, the production of lactic acid as a by-product was effectively minimized. This approach enabled *Lactobacillus casei* to multiply rapidly to high concentrations, laying a foundation for the industrial production of high-yield aroma starters. This advancement supports the bacterium's application in various sectors, including dairy processing and functional food production.

**KEYWORDS** *Lacticaseibacillus casei* TCS, high-density fermentation, medium optimization, ANN-GA, ion exchange resin, fed-batch fermentation

*L*acticaseibacillus *casei* TCS, a Gram-positive, non-spore-forming bacterium, is extensively studied because of its important applications in the food industry. In our previous study, we engineered recombinant *L. casei* strains by expressing NADH oxidase, acetolactate synthase, and inactivated acetolactate decarboxylase in *L. casei* TCS. These strains, when added to yogurt, increased the production of diacetyl and acetoin (1). *L. casei* TCS rapidly reproduces under anaerobic or facultative anaerobic conditions, requiring a nutrient-rich environment and specific growth factors for optimal growth. Moreover, it exhibits notable metabolic activities, including acid, aroma,

**Peer Reviewers** Rajib Saha, University of Nebraska-Lincoln, Lincoln, Nebraska, USA; Zhihong Sun, Inner Mongolia Agricultural University, Inner Mongolia, China

Address correspondence to Chang Ge, gechang@sit.edu.cn.

The authors declare no conflict of interest.

and exopolysaccharide production; protein degradation; and antibacterial capabilities. These characteristics make *L. casei* valuable for producing fermented dairy products (e.g., yogurt, cheese, and milk wine), lactic acid, bacteriocins, enzymes, vitamins, and probiotics (2).

Lactic acid bacteria (LAB), including *L. casei*, play a crucial role in the fermentation process by producing flavor compounds through carbohydrate metabolism and protein and fat hydrolysis. These processes result in the formation of various volatile organic compounds and lactic acid, enhancing the flavor profiles of fermented dairy products. Moreover, *L. casei* improves human health by promoting gut microbiota diversity, increasing protein digestibility, and enhancing the nutritional value of food proteins. LABs such as *L. casei* are widely used as starter cultures in the food processing industry. They help achieve desirable product characteristics by fermenting substrates and producing compounds beneficial for food safety and human health (3).

Optimizing culture media and fermentation conditions is crucial for maximizing the industrial applications of *L. casei*. Although the traditional de Man, Rogosa, and Sharpe (MRS) medium is effective, it is expensive and contains complex ingredients such as peptone, beef extract, and yeast extract, which are not ideal for large-scale production. Therefore, developing an economical and suitable food-grade medium that supports efficient fermentation is essential for industrial applications. Recent studies have focused on optimizing various factors such as culture medium composition (carbon and nitrogen sources, growth factors, and inorganic salts), culture conditions (temperature, pH, and dissolved oxygen), and incubation periods. These factors significantly affect microbial fermentation, and their optimization is a primary method for enhancing fermentation efficiency (4–7). In addition, agricultural waste, which contains abundant nutrients, has been explored as a sustainable and cost-effective substrate for lactic acid production. This approach not only reduces production costs but also helps address environmental concerns. The limitations of traditional optimization methods under conditions of limited sample data are evident in their inability to effectively identify the global optimum in the absence of comprehensive global information (8). For instance, the effectiveness and limitations of the response surface methodology (RSM) in the simulation of discrete and stochastic manufacturing systems have been evaluated. A main disadvantage of RSM is its inability to consider interactive effects among variables, which is crucial for accurately determining output-input relationships (9). Meanwhile, artificial neural network–genetic algorithm (ANN–GA) is highly effective for solving complex, non-linear, and high-dimensional problems. ANNs model intricate input-output relationships without explicit formulas, while GA explores vast search spaces to identify global optima, overcoming local optima issues inherent in traditional methods (10). Additionally, ANN–GA's parallelizability enhances scalability, making it suitable for large-scale or real-time optimization tasks, offering superior performance in dynamic and uncertain environments.

Batch fermentation is simple to operate and has a low risk of contamination, but it often suffers from low productivity due to substrate and product inhibition (11). In contrast, fed-batch fermentation allows for the continuous addition of substrates, which helps maintain optimal nutrient levels and achieve high biomass yields. This method effectively mitigates the inhibitory effects of high substrate concentrations and end-product accumulation, making it preferable for high-density cultures (12–14). The resin feed system introduces an innovative mechanism for selective lactate removal, addressing the pH drop and cell growth inhibition caused by lactate accumulation through the adsorption of lactate molecules (15). In distinction from conventional pH regulation approaches and methods, such as alkali neutralization, this system effectively removes lactic acid while avoiding excessive addition of alkaline substances, maintaining fermentation stability, and minimizing by-product interference (16). Its dynamic regulation and adaptive adjustment capabilities ensure lactate concentration remains within an optimal range, enhancing fermentation efficiency and preventing inhibition

(17). Therefore, selectively removing lactic acid during the fermentation process is crucial for maintaining high cell densities and improving overall fermentation efficiency (18, 19).

In this study, we used a combination of single-factor and orthogonal experimental designs and the response surface methodology to optimize the medium composition for *L. casei* TCS (20, 21). In addition, we developed kinetic models for cell growth and substrate consumption by using logistic formulas and the Luedeking–Piret model, respectively. The high determination coefficient ($R^2$) values obtained using these models confirmed the accurate simulation of *L. casei* TCS growth and substrate consumption (22, 23). Furthermore, we created a high-efficiency production process that integrates resin absorption (24) and fed-batch operation, significantly enhancing cell density (25) and fermentation efficiency (26, 27). Optimization of the fermentation process not only enhances the yield (28, 29) and quality of end products but also reduces production costs (30), making it economically feasible for large-scale applications. Given the increasing demand for fermented products and probiotics, optimizing the fermentation process of *L. casei* TCS is highly promising for industrial applications in the food and health sectors (31).

## MATERIALS AND METHODS

### Microorganism and culture conditions

*L. casei* TCS was isolated from traditionally fermented yogurt by the Food Flavor Control Innovation Team at the School of Flavor Technology and Engineering, Shanghai University of Applied Technology, China. The isolated *L. casei* TCS was stored in MRS broth with 20% (vol/vol) glycerol at −80℃ and preserved at the typical culture preservation center of China, located at Wuhan University, Wuhan City, Hubei Province, China, 430072, under the preservation number CCTCC no. M2017364. For activation, the preserved *L. casei* TCS was inoculated in sterilized MRS medium at a 2% (vol/vol) concentration, cultured at 37℃ for 12 h, and activated twice to produce active strains that could be used as seed liquid for subsequent experiments.

The seed medium was composed of 20 g/L of glucose, 10 g/L of peptone, 10 g/L of beef extract, 5 g/L of yeast extract, 2 g/L of $K_2HPO_4$, 1 g/L of Tween 80, 5 g/L of sodium acetate, 2 g/L of ammonium citrate, 2 g/L of dipotassium hydrogen phosphate, 2 g/L of diammonium hydrogen citrate, 0.3 g/L of $MgSO_4$, and 0.2 g/L of $MnSO_4$. The pH of the medium was adjusted to 6.2–6.4 before autoclaving it at 121℃ for 15 min.

For the experiments, apples, carrots, tomatoes, sweet corn, and oyster mushrooms were purchased from the market. The juices were extracted by stirring the chopped fruits and vegetables with water and then filtering the mixture through gauze to obtain juices from apples, carrots, tomatoes, sweet corn, and oyster mushrooms.

### Optimization of medium composition

The composition of the basic MRS medium was optimized by adjusting the carbon source, nitrogen source, and buffer salt composition and by adding growth factors. The culture conditions were standardized as follows: inoculation amount of 2%, culture temperature of 37℃, and culture duration of 12 h.

#### *Carbon source*

The base medium was based on the MRS formulation and included 10 g/L of peptone, 10 g/L of beef extract, 5 g/L of yeast extract, 5 g/L of sodium acetate, 2 g/L of ammonium citrate, 2 g/L of dipotassium hydrogen phosphate, 1 g/L of Tween 80, 0.3 g/L of anhydrous magnesium sulfate, and 0.2 g/L of manganese sulfate monohydrate. We investigated the effects of various carbon sources, namely glucose, galactose, glucose:galactose (3:1), sucrose, sorbitol, and glucose:sorbitol (3:1), on the number of viable bacteria in the fermentation broth.

### Nitrogen source

Following the optimization of carbon sources (5 g/L of sodium acetate, 2 g/L of ammonium citrate, 2 g/L of dipotassium hydrogen phosphate, 1 g/L of Tween-80, 0.3 g/L of anhydrous magnesium sulfate, and 0.2 g/L of manganese sulfate monohydrate), we investigated the effects of different nitrogen sources (peptone, skim milk, yeast extract, beef extract, acid hydrolyzed casein, and serine; each at 20 g/L) on the number of viable bacteria in the fermentation broth.

### Growth factors

Based on the results of the optimization of carbon and nitrogen sources, we used a base medium consisting of 5 g/L of sodium acetate, 2 g/L of ammonium citrate, 2 g/L of dipotassium hydrogen phosphate, 1 g/L of Tween-80, 0.3 g/L of anhydrous magnesium sulfate, and 0.2 g/L of manganese sulfate monohydrate to investigate the effects of different growth factors (apple, carrot, tomato, sweet corn, and *Pleurotus ostreatus* juices, each at 20 g/L) on the number of viable bacteria in the fermentation broth. In addition, we determined the optimal concentrations of the most effective growth factor identified in the initial tests. The concentrations tested were 5, 10, 20, and 40 g/L, and we examined their effects on the number of viable bacteria in the fermentation broth.

### Buffer salt combination

Based on the results of the optimization of carbon and nitrogen sources and growth factors (g/L), we used the base medium consisting of 1 g/L of Tween 80, 0.3 g/L of anhydrous magnesium sulfate, and 0.2 g/L of manganese sulfate monohydrate. We examined the effects of various buffer salt combinations (5 g/L of sodium acetate, 2 g/L of ammonium citrate, and 3 g/L of dipotassium hydrogen phosphate as the central level, with each factor set at three levels) on the number of viable bacteria in fermentation broth by using an orthogonal design. The sodium acetate concentrations were set at 2, 5, and 8 g/L, while the ammonium citrate and potassium dihydrogen phosphate concentrations were adjusted to 1, 2, and 3 g/L, respectively. To ensure the applicability of analysis of variance (ANOVA), Shapiro-Wilk and Levene's tests were conducted to verify the assumptions of normality and homogeneity of variances.

### Artificial neural network model establishment and genetic algorithm optimization design

Based on the optimization results of single-factor and orthogonal tests, the appropriate carbon-to-nitrogen ratio was adjusted experimentally. The central levels and range intervals for the three components—carbon source (glucose), nitrogen source (peptone), and growth factor (sweet corn juice)—were determined using a Box–Behnken design. The levels were set at −1, 0, and 1, corresponding to glucose concentrations of 10, 20, and 30 g/L; peptone concentrations of 20, 40, and 60 g/L; and sweet corn juice concentrations of 4, 8, and 12 g/L, respectively. In this experiment, the medium consisted of 1 g/L of Tween 80, 0.3 g/L of anhydrous magnesium sulfate, and 0.2 g/L of manganese sulfate monohydrate.

   An ANN model was constructed using the concentrations of glucose, peptone, and sweet corn juice as the input layer and the number of viable bacteria as the output layer. We modified the number of neurons in the hidden layer through multiple training sessions to select a minimal number of hidden layers and prevent overfitting. The data set was divided into 70% for training, 15% for validation, and 15% for testing. We compared the modeling effects of different training algorithms to select the most appropriate one. The training continued until the ANN model met the predetermined training stop criteria. The fitting and prediction accuracies of the model were evaluated using $R^2$, mean absolute error (MAE), and root mean square error (RMSE). To ensure the stability of the model, a fivefold (k = 5) cross-validation was employed for model evaluation in this study. GA was used to optimize the constructed ANN model globally,

and the optimal combination of media components to maximize the number of viable bacteria in the fermentation broth was studied. Optimized for GA, the objective function (maximizing the number of viable bacteria) was converted to the GA fitness function to minimize the negative values of the objective function.

In terms of GA parameter setting, the number of variables is set to 3, the population size is set to 50, the maximum number of iterations is set to 300, the crossover probability is set to 0.8, and the proportion of elite individuals is set to 0.05. The initial population is generated by the stochastic uniformity method, and the adaptive mutation operator is used to enhance the search ability. In addition, the stopping criterion is based on the fitness rate change threshold, which stops iterations when the fitness value changes less than $10^{-6}$ for 20 consecutive generations to ensure convergence and avoid unnecessary computational overhead.

In order to optimize the GA parameters, we adjusted the population size (30–100), crossover probability (0.6–0.9), and maximum number of iterations (100–500), respectively, and selected the final parameter combination based on the convergence speed and stability of the objective function. At the same time, the robustness of the GA is evaluated by multiple independent runs to ensure the reliability of the optimization results.

## Determination of lactic acid content

The method used to determine the lactic acid content in the fermentation supernatant mainly followed the acid–base indicator titration technique outlined in GB12456-2021, where total acid is quantified as lactic acid. The operation steps were as follows: 2 mL of the fermentation supernatant was mixed with 5 mL of distilled water and two drops of phenolphthalein indicator in a small conical flask. This mixture was titrated with a 0.01 mol/L sodium hydroxide standard solution until it turned slightly red and the color persisted for 30 s. A blank test was also performed. Each experimental condition was repeated three times, and the average volume of sodium hydroxide standard titration solution consumed was recorded. The formula (equation 1) used to calculate the total acid content is as follows:

$$P = \frac{c \times (V_1 - V_0) \times k}{V} \times 1000, \tag{1}$$

where $P$ is the total acid content (g/L), $c$ is the concentration of sodium hydroxide standard titration solution (mol/L), $V_1$ is the volume of sodium hydroxide standard titration solution consumed when titrating the fermentation supernatant (mL), $V_0$ is the volume of the sodium hydroxide standard titration solution consumed when titrating the blank sample (mL), $k$ is the conversion coefficient for lactic acid (0.090), $V$ is the volume of the fermentation supernatant (mL), and 1,000 is the conversion factor used to express the acid content in grams per liter. Each experimental setup was replicated three times, and the average value of these replicates was used to ensure the reliability of the results.

## Determination of glucose content

The glucose content in the fermentation supernatant was quantified using the 3,5-dinitrosalicylic acid (DNS) colorimetric method, as outlined in NY/T 2742-2015. This standardized method is recognized for its accuracy in measuring soluble sugars in various matrices, including fermentation broths.

The procedure involved two main stages: the preparation of a glucose standard curve and the measurement of glucose concentrations in samples. For the standard curve, glucose standard solutions (1 mg/mL) were prepared in volumes ranging from 0 to 1.0 mL and adjusted to a final volume of 2 mL with distilled water in stoppered test tubes. Each tube then received 2 mL of DNS reagent, which had been prepared a week earlier. The mixtures were boiled for 5 min in a water bath and then rapidly cooled to room temperature. The solutions were diluted to 10 mL, and their absorbance

was measured at 540 nm using an ultraviolet spectrophotometer to create a standard curve, where glucose concentration was plotted on the abscissa and absorbance on the ordinate. In the second stage, the same procedure was applied to the fermentation supernatant samples. The glucose concentration in these samples was determined by comparing their absorbance readings against the standard curve. This method ensured the precise and reliable quantification of glucose, which is essential for evaluating the fermentation efficiency of *L. casei* TCS.

## Fermentation kinetics models

The fermentation kinetics of *L. casei*, a homofermentative lactic acid bacterium, involves cell growth, substrate consumption, and product formation kinetics (32). During fermentation, one molecule of glucose is converted into two molecules of lactic acid, with a conversion efficiency exceeding 90% (12). This high conversion efficiency highlights the importance of accurately modeling the growth of *L. casei*, its glucose utilization, and lactic acid production.

Cell growth kinetics model:

The logistic growth model is suitable for *L. casei*, reflecting its rapid growth and high bacterial concentration (33, 34). The model is defined as follows:

$$\frac{dX}{dt} = \mu_{max} X \left( 1 - \frac{X}{X_{max}} \right). \tag{2}$$

In this formula, $X$ represents the viable bacterial count (lg CFU/mL), $\mu_{max}$ is the maximum growth rate (h$^{-1}$), $X_{max}$ denotes the carrying capacity, and $t$ is the culture time in hours. The integration of this formula and substitution of $X = X_0$ (the initial bacterial count at $t = 0$) leads to

$$X(t) = \frac{X_0 e^{\mu_{max} t}}{1 - \frac{X_0}{X_{max}} \left( 1 - e^{\mu_{max} t} \right)}. \tag{3}$$

To accommodate non-zero initial conditions, the model is modified as follows:

$$X(t) = \frac{a e^{ct}}{1 - \frac{a}{b} \left( 1 - e^{ct} \right)} + d. \tag{4}$$

Where $a = X_0$, $b = X_{max}$, $c = \mu_{max}$, since equation (3) represents the change in biomass increment, while the initial biomass value is not zero, so $d$ corrects for the initial bacterial load.

To establish this model, the values of $t$ and $X$ are substituted into the formula. Using MATLAB's curve fitting app, we selected a custom formula (equation 4) for non-linear fitting to determine the values of $a$, $b$, $c$, and $d$. This process yielded the functional expression of the bacterial growth kinetics model. The data from the fitted curve were extracted, and the growth model curve was plotted.

Product formation kinetics model:

The Luedeking–Piret model, which indicates a correlation between lactic acid production and bacterial growth, is expressed as follows:

$$\frac{dP}{dt} = m \frac{dX}{dt} + nX. \tag{5}$$

In this model, $P$ represents the lactic acid concentration (g/L), and $m$ and $n$ are empirical constants. This model was integrated with the logistic growth model to obtain insights into the dynamics of lactic acid production.

Then, equation 3 was substituted into equation 5 to provide a functional expression for lactic acid production over time:

$$P(t) = P_0 + m\frac{ae^{ct}}{1 - \frac{a}{b}(1 - e^{ct})} + n\frac{b}{c}\ln\left[1 - \frac{a}{b}(1 - e^{ct})\right]. \tag{6}$$

In this expression, $S_0$ represents the initial lactic acid concentration. The parameters $a$, $b$, $c$, $m$, $n$, and $P_0$ were determined by fitting $t$ and $P$ values, using MATLAB's curve fitting app with equation 6.

Substrate consumption kinetics model:

A modified Luedeking–Piret model was used to evaluate glucose consumption, relating substrate use to cell growth:

$$\frac{dS}{dt} = -M\frac{dX}{dt} - NX. \tag{7}$$

In this model, $S$ represents the glucose content (g/L), $M$ is an empirical constant related to the growth rate, and $N$ is another empirical constant related to cell concentration. This model, denoted as equation 7, illustrates how the glucose consumption rate is affected by both the rate of bacterial growth and the concentration of bacteria.

Equation 4 was substituted into equation 7 to provide a detailed functional expression of glucose concentration over time as follows:

$$S(t) = S_0 - M\frac{ae^{ct}}{1 - \frac{a}{b}(1 - e^{ct})} - N\frac{b}{c}\ln\left[1 - \frac{a}{b}(1 - e^{ct})\right]. \tag{8}$$

In this expression, $S_0$ represents the initial glucose concentration. Parameters $a$, $b$, $c$, $M$, $N$, and $S_0$ were determined by fitting $t$ and $S$ values using MATLAB's curve fitting app with equation 8.

## Pretreatment of weakly basic anion exchange resin

Amberlite IRA-67, a weakly basic anion exchange resin made from cross-linked acrylate polymer with particle sizes of 0.5–0.75 mm, was used for the adsorption treatment of organic acids, such as lactic acid. This resin was reported to be more effective than strongly basic anion exchange resins for this purpose (35). The resin pretreatment process involved several steps: (i) the new resin was initially soaked in deionized water. (ii) It was then immersed in 1 mol/L sodium hydroxide solution at five times the resin's volume for 1 h. This step converted the resin into its OH form, achieved through repeated stirring with a glass rod. (iii) The resin was washed with distilled water (10 times the resin's volume) until the effluent reached a neutral pH. (iv) The resin was treated in 1 mol/L hydrochloric acid solution at five times the resin's volume for 1 h. This step converted the resin into its Cl form, achieved through repeated stirring with a glass rod. (v) Finally, the resin was soaked in distilled water and sealed for later use.

## Suppressing lactic acid inhibition using ion exchange

During the fermentation process, the prepared seed liquid was inoculated into a 5 L fermentation tank containing 2.5 L of optimized medium. Amberlite IRA 67, a strongly basic cation-exchange resin, was utilized in the system. The flow rate was set at 0.5, 1, and 1.5 BV/h. The tank conditions were set to a rotation speed of 90 r/min and a temperature of 37℃, without ventilation or pH control. During fermentation, particularly when the culture reached the mid-logarithmic growth phase or the pH dropped to 5.0, sterilized resin (prepared as described earlier) was added under flame protection through the inoculation port. This addition was carefully monitored to maintain the pH between 5.0 and 5.5. Every 2 h, a sample of the fermentation broth was collected; 1 mL was immediately used for measuring the viable bacterial count, and the remainder was centrifuged at 8,000 r/min for 10 min. The supernatant was then stored at −20℃ for subsequent analysis of residual glucose content.

## Relief of substrate inhibition by fed-batch method

Following the addition of the resin and the onset of the feeding stage, a concentrated feeding liquid was introduced through the inoculation port under flame protection. This feeding solution primarily consisted of glucose (200 g/L), peptone (150 g/L), and a small quantity of ammonium citrate (10 g/L), which served as the carbon and nitrogen sources.

## Statistical analysis

Each experiment was performed in triplicate to ensure the reliability of the results. Average values and standard deviations were calculated. For statistical analysis, ANOVA was performed, and Tukey's test was used for multiple comparisons. Significance levels were set at $P < 0.05$ and $P < 0.01$. All data analyses and graph plotting were performed using SPSS 21.0 and Origin 2018.

## RESULTS AND DISCUSSION

### Carbon and nitrogen source optimization results

Our findings demonstrated that LAB could more easily use monosaccharides than polysaccharides. The growth of *L. casei* TCS was significantly better in the presence of glucose than in the presence of other carbon sources (Fig. 1a). Glucose, a monosaccharide, was directly used by LAB, significantly enhancing their growth and metabolism. The number of viable bacteria was slightly higher in fermentation experiments using a mixture of two carbon sources, with one being dominant, than in fermentation experiments using a single carbon source; however, this difference was not statistically significant. This finding indicated that although mixed carbon sources were beneficial, glucose alone effectively promoted growth. Thus, glucose was selected as the carbon source in the optimized medium.

Further refinement of the medium involved optimizing the nitrogen source while maintaining a constant glucose level of 20 g/L. The basic MRS medium was used as the control in this experiment. As shown in Fig. 1b, the growth of *L. casei* TCS was superior

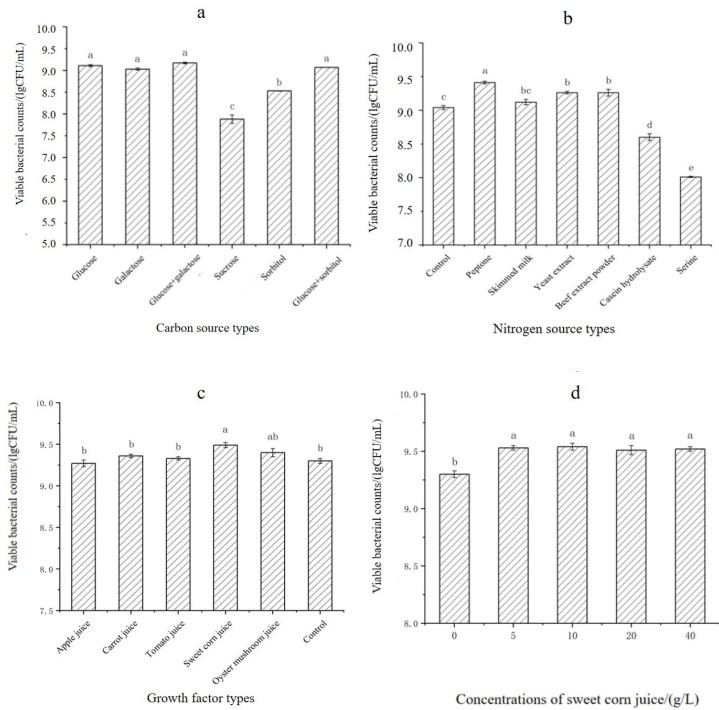

**FIG 1** Effects of carbon sources (a), nitrogen sources (b), growth factors (c), and different sweet corn juice concentrations (d) on the number of viable bacteria in the fermentation broth.

in the medium containing peptone, skim milk, yeast extract, and beef extract compared with the control medium. Peptone significantly outperformed other nitrogen sources. Conversely, serine was less effective in supporting growth due to its simple composition. On the basis of these findings and our goal to streamline the medium's complexity, peptone was selected as the nitrogen source for the culture medium.

## Growth factor optimization results

### *Selection of growth factors*

Under controlled conditions with glucose and peptone, various growth factors were added to the medium. We investigated the ability of fruit and vegetable juices, which are cost-effective and widely available, to promote bacterial growth. Sweet corn juice demonstrated the best enrichment effect, significantly outperforming other growth factors (Fig. 1c). The number of viable bacteria obtained with sweet corn juice was 2.38 times higher than that obtained without any growth factors. This finding confirmed the effectiveness of sweet corn juice as a growth factor for optimizing the medium.

### *Concentration of sweet corn juice*

Different concentrations of sweet corn juice (5, 10, 20, and 40 g/L) were examined to determine the optimal concentration. As shown in Fig. 1d, the enrichment effect did not significantly differ among these concentrations. Therefore, the smallest concentration of 5 g/L was selected for subsequent experiments to simplify the medium composition and reduce costs.

## Optimization results and analysis of buffer salts

Based on the optimized carbon and nitrogen sources as well as growth factors, the concentration combinations of three buffer salts (sodium acetate, ammonium citrate, and dipotassium hydrogen phosphate) were optimized using an orthogonal design. These buffer salts play a crucial role in maintaining pH stability during fermentation, which is essential for the growth and metabolic activities of *L. casei* TCS. We used the orthogonal design to determine the optimal combination of buffer salts, with each factor set at three levels. The design matrix and corresponding viable bacterial counts are presented in Table 1. ANOVA was performed to identify the primary and secondary factors affecting viable bacterial counts (Table 2). ANOVA was performed to identify the primary and secondary factors affecting viable bacterial counts (Table 2). The analysis revealed that ammonium citrate had the most significant effect on viable bacterial counts (F = 20.26, $P = 0.047$, $\eta^2 = 0.483$), followed by dipotassium hydrogen phosphate (F = 15.65, $P = 0.060$, $\eta^2 = 0.373$), whereas sodium acetate had a relatively smaller impact (F = 5.05, $P = 0.165$, $\eta^2 = 0.120$). The 95% confidence interval for viable bacterial counts was [9.35, 9.46]. Post hoc comparisons using Tukey's test indicated significant differences among ammonium citrate levels ($P < 0.05$). The optimal buffer

**TABLE 1** Orthogonal test design and test results of buffer salt combinations

| Serial number | Sodium acetate | Ammonium citrate | Potassium dihydrogen phosphate | Logarithm of viable bacterial counts (lg CFU/mL) |
|:---:|:---:|:---:|:---:|:---:|
| 1 | 1 | 1 | 1 | 9.32 ± 0.04 |
| 2 | 1 | 2 | 2 | 9.43 ± 0.02 |
| 3 | 1 | 3 | 3 | 9.39 ± 0.02 |
| 4 | 2 | 1 | 2 | 9.38 ± 0.02 |
| 5 | 2 | 2 | 3 | 9.49 ± 0.05 |
| 6 | 2 | 3 | 1 | 9.30 ± 0.07 |
| 7 | 3 | 1 | 3 | 9.49 ± 0.07 |
| 8 | 3 | 2 | 1 | 9.46 ± 0.03 |
| 9 | 3 | 3 | 2 | 9.35 ± 0.01 |

**TABLE 2** Three-factor analysis of variance results[c]

| Source of variance | Quadratic sum | df | Mean square | F | P |
|---|---|---|---|---|---|
| Intercept | 795.428 | 1 | 795.428 | 1,664,849.326 | 0.000[b] |
| Sodium acetate | 0.005 | 2 | 0.002 | 5.047 | 0.165 |
| Ammonium citrate | 0.019 | 2 | 0.010 | 20.256 | 0.047[a] |
| Dipotassium phosphate | 0.015 | 2 | 0.007 | 15.651 | 0.060 |
| Residual error | 0.001 | 2 | 0.000 | /[d] | / |

[a]$P < 0.05$ significant.
[b]$P < 0.001$ extremely significant.
[c]$R^2$: 0.976.
[d]"/", Not available.

salt combination (2 g/L of ammonium citrate, 3 g/L of dipotassium hydrogen phosphate, and 5 g/L of sodium acetate) resulted in the highest viable bacterial count of 9.49 lg CFU/mL. These results indicate that ammonium citrate significantly enhances the buffering capacity of the medium, thereby maintaining a stable pH and supporting bacterial growth. The selected buffer salts effectively maintained the pH within the optimal range, preventing fluctuations that could inhibit bacterial growth and metabolism. The stability of pH during fermentation is crucial for maximizing the growth of *L. casei* TCS. Overall, the optimized buffer salt combination provides a balanced environment that supports the high-density fermentation of *L. casei* TCS, facilitating efficient production processes for industrial applications.

## ANN fitting model and GA optimization analysis

Based on the results of medium composition and concentration optimization, the content of trace components in the medium was determined as follows: 5 g/L of sodium acetate, 2 g/L of ammonium citrate, 3 g/L of dipotassium hydrogen phosphate, 1 g/L of Tween 80, 0.3 g/L of anhydrous magnesium sulfate, and 0.2 g/L of manganese sulfate monohydrate. The concentrations of buffer salts were optimized as described previously. Tween 80 was included as an emulsifier to uniformly disperse cells. Magnesium and manganese ions, which are essential trace elements for the growth of LAB, were added in small amounts based on reference concentrations from the MRS medium.

The concentrations of glucose, peptone, and sweet corn juice were further optimized using a three-factor, three-level Box–Behnken design (Table 2). The specific test plan and results are shown in Fig. 2.

Using the Box–Behnken test data, we established and trained an ANN model with the neural network toolbox in MATLAB R2018a. The training algorithm and the number of neurons in the hidden layer varied across multiple training sessions. The best training effect was observed when using the Bayesian regularization algorithm with three hidden layers, resulting in a 3-3-1 topology (Fig. 3a). Each hidden layer employed the tansig (hyperbolic tangent sigmoid) activation function, while the output layer used a purelin (linear) activation function to ensure smooth regression predictions. To prevent overfitting while ensuring training accuracy, the model was precisely fit to the test data. As shown in Fig. 3b, the RMSE of the training reached $5.9327 \times 10^{-5}$ at the 86th iteration, demonstrating rapid convergence. The fitting effect of ANN training is presented in Fig. 3c. After training, the $R^2$ of the ANN model reached 0.9656, MAE: 0.01, indicating a strong fitting ability and accurate prediction, which are suitable for further optimization. The experimental results of K-fold cross-validation (k = 5) are as follows: training set RMSE (mean): $5.9196 \times 10^{-5}$, $R^2$ (mean): 0.9655, MAE (mean): 0.0100; validation set RMSE (mean): $6.0862 \times 10^{-5}$, $R^2$ (mean): 0.9621, MAE (mean): 0.0106; testing set RMSE (mean): $6.1244 \times 10^{-5}$, $R^2$ (mean): 0.9617, MAE (mean): 0.0110. The low RMSE values in both the training and validation sets suggest that the model exhibits stable training behavior and possesses good generalization capabilities. The $R^2$ values of approximately 0.9655 for the training set and 0.9617 for the test set indicate that the model fits the data well without significant overfitting. The MAE of 0.0110 for the test set suggests a small prediction

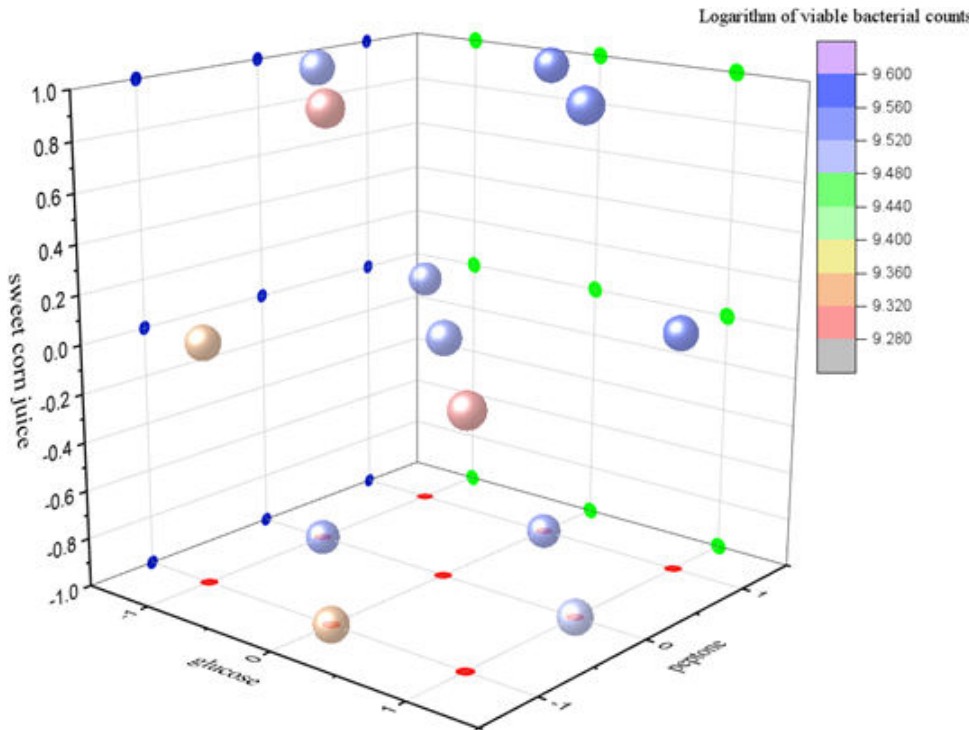

**FIG 2** Box-Behnken design and results.

error, implying high prediction accuracy. Furthermore, the consistency in cross-validation results, with minimal fluctuation in errors across different folds during training, validation, and testing, underscores the stability and reliability of the model.

To determine the optimal values for the established model, we used a GA. The function file of the ANN model was transformed into a fitness function to determine the minimum value. By adjusting the value interval of independent variables and combining the algorithm with actual test conditions, we identified the optimal solution within a certain range. The GA optimization process is illustrated in Fig. 4a. Because the GA targets the minimum value of the function, the fitness function was subjected to a linear transformation of "f = −f." The optimal value corresponding to the number of viable

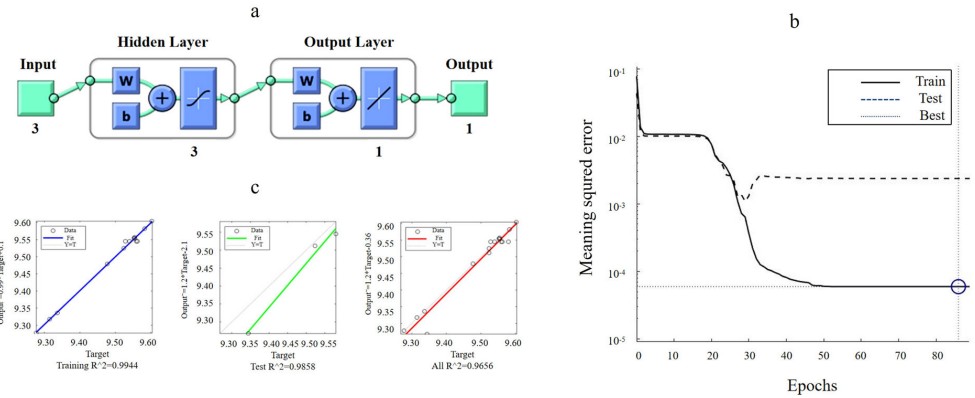

**FIG 3** ANN structure (a) and training performance (b) figure. The selection is made for three hidden layers. The result of the training mean squared error is shown as the solid line; the result of the testing mean squared error is shown as the dashed line; the best training mean squared error is shown as the dotted line. Regression and fitting figures for ANN training (c). The sample data were shown as a hollow circle. The fitting effect was shown as the solid line in color; the value of the ordinate is represented by Y; the value of the horizontal coordinate is represented by T.

bacteria was 9.66 lg CFU/mL, achieved using a combination of 35 g/L of glucose, 62 g/L of peptone, and 24 g/L of sweet corn juice.

Finally, we verified the optimal solution predicted by the ANN–GA. The actual number of viable bacteria determined was 9.66 lg CFU/mL, which exactly matched the predicted value. After optimization, the number of viable cells increased by 4.2 times compared with that obtained using the basic MRS medium (9.04 lg CFU/mL). These results demonstrated that ANN–GA optimization resulted in a higher number of viable bacteria, with verification tests confirming the reliability of the predictions. The results demonstrated that the ANN–GA exhibited strong fitting and prediction abilities, making it suitable for optimizing microbial fermentation media.

## Kinetic models of cell growth

Table 3 presents the number of viable bacteria and concentrations of lactic acid and glucose at different fermentation stages during the batch culture of *L. casei* TCS. After 10 h, cell growth entered a stable phase, and the highest number of viable bacteria was observed at approximately 12 h. At this point, the fermentation was nearly complete, indicating that the number of viable bacteria would not increase further. To determine the cell growth pattern, the number of viable bacteria at different fermentation stages was non-linearly fitted, and a kinetic model of cell growth was established using MATLAB.

To fit the data, we used the curve fitting app in MATLAB and selected time (t) and viable bacterial counts (X) as the independent and dependent variables, respectively. Custom equation 4 was used for fitting. The coefficient constraints for the fitting options are shown in Table 4.

After adjustment, the fitting curve (Fig. 4b) exhibited a good fit to the experimental data, with an $R^2$ of 0.9998 and an RMSE of 0.0156, accurately reflecting the actual growth of the bacteria. The coefficients (95% confidence intervals) were determined as follows: $a$ = 0.04773 (0.0299, 0.06555), $b$ = 2.106 (2.039, 2.172), $c$ = 0.683 (0.6247, 0.7412), and $d$ = 7.582 (7.532, 7.632).

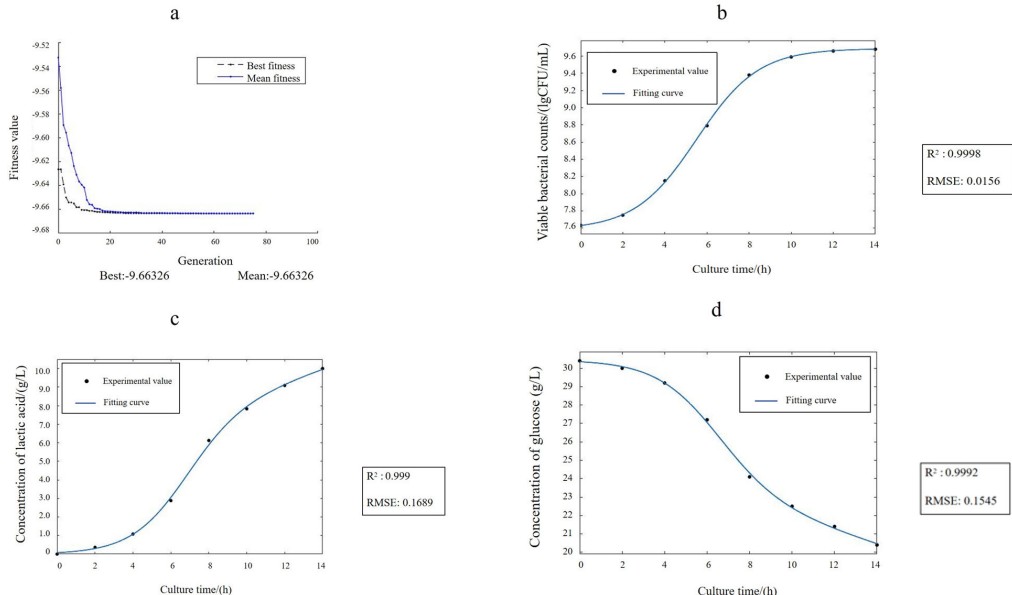

**FIG 4** GA optimization process (a). The value of the mean fitness was shown as the solid line. The value of the best fitness is shown as the dashed line. Changes in viable bacterial counts over time and the kinetic fitting model (b). The result of the experimental value was shown as a solid circle; the result of the fitting curve was shown as a solid circle. Changes in lactic acid content over time and the kinetic fitting model (c). The result of the experimental value was shown as a solid circle; the result of the fitting curve was shown as a solid circle. Changes in glucose content over time and the kinetic fitting model (d). The result of the experimental value was shown as a solid circle; the result of the fitting curve was shown as a solid circle.

**TABLE 3** Number of viable bacteria and concentrations of lactic acid and glucose at different stages of batch culture

| Culture time (h) | Logarithm of viable bacterial counts (lg CFU/mL) | Lactic acid content (g/L) | Glucose content (g/L) |
|---|---|---|---|
| 0 | 7.63 ± 0.00 | 0 ± 0.00 | 0.00 ± 0.00 |
| 2 | 7.75 ± 0.01 | 0.36 ± 0.02 | 0.40 ± 0.01 |
| 4 | 8.15 ± 0.02 | 1.08 ± 0.03 | 1.20 ± 0.02 |
| 6 | 8.79 ± 0.01 | 2.88 ± 0.04 | 3.20 ± 0.01 |
| 8 | 9.38 ± 0.03 | 6.12 ± 0.07 | 6.30 ± 0.03 |
| 10 | 9.59 ± 0.02 | 7.83 ± 0.08 | 7.90 ± 0.02 |
| 12 | 9.66 ± 0.02 | 9.09 ± 0.14 | 9.00 ± 0.05 |
| 14 | 9.68 ± 0.04 | 9.99 ± 0.07 | 10.10 ± 0.04 |

By substituting these values into the equation, we expressed the bacterial growth kinetic model (equation 9) as follows:

$$X(t) = \frac{0.04773e^{0.683t}}{1 - \frac{0.04773}{2.106}(1 - e^{0.683t})} + 7.582. \tag{9}$$

This model provides a robust description of the growth kinetics of *L. casei* TCS during batch culture, offering valuable insights for optimizing fermentation processes.

## Kinetic model of product formation

To elucidate the patterns of lactic acid production, we non-linearly fitted the lactic acid concentration at different fermentation stages and established a kinetic model of lactic acid production using MATLAB. By controlling the initial value of the bacteria and maintaining a constant specific growth rate, we determined coefficient constraints for the model (Table 5).

We conducted the fitting process using these constraints. After adjustment, the fitting curve was obtained (Fig. 4c). The fitting results were highly accurate, with an $R^2$ value of 0.9990 and an RMSE of 0.1689, effectively reflecting the actual lactic acid production by the bacteria. The coefficients (95% confidence intervals) were determined as follows: $P_0$ = 2.367 × 10$^{-14}$ (fixed interval), $a$ = 0.04772 (−1.729e + 05, 1.729e + 05), $b$ = 4.821 (−1.746e + 07, 1.746e + 07), $c$ = 0.683 (fixed interval), $m$ = 1.559 (−5.646e + 06, 5.646e + 06), and $n$ = 0.07135 (−2.584e + 05, 2.584e + 05).

By substituting these values into equation 6, we expressed the kinetic model (equation 10) for lactic acid production as follows:

$$P(t) = 1.559\frac{0.04772e^{0.683t}}{1 - \frac{0.04772}{4.821}(1 - e^{0.683t})} + 0.07135\frac{4.821}{0.683}\ln\left[1 - \frac{0.04772}{4.821}(1 - e^{0.683t})\right]. \tag{10}$$

This model provides a robust description of the kinetics of lactic acid production during the batch culture of *L. casei* TCS. The high $R^2$ and low RMSE values indicate the reliability and accuracy of the model in predicting lactic acid production, which is essential for optimizing fermentation processes.

**TABLE 4** Coefficient constraint conditions of the biomass growth kinetic model

| Coefficient | Initialization value | Lower limit | Upper limit |
|---|---|---|---|
| *a* | 0.0500 | 0 | Inf |
| *b* | 2.0000 | 0 | Inf |
| *c* | 0.5000 | 0 | Inf |
| *d* | 7.6000 | 0 | Inf |

**TABLE 5** Coefficient constraint conditions of the product generation kinetic model

| Coefficient | Initialization value | Lower limit | Upper limit |
| --- | --- | --- | --- |
| $P_0$ | 0.0000 | 0 | Inf |
| $a$ | 0.0477 | 0.0477 | 0.0477 |
| $b$ | 2.0000 | 2 | 10 |
| $c$ | 0.6830 | 0.6830 | 0.6830 |
| $m$ | 0.9000 | −Inf | Inf |
| $n$ | 0.0200 | −Inf | Inf |

## Kinetic model of substrate consumption

To elucidate the pattern of glucose consumption during fermentation, we non-linearly fitted the glucose concentration at different fermentation stages and established a kinetic model for glucose consumption using MATLAB. The initial values of the bacteria and the specific growth rate were kept constant. The coefficient constraints for the model are listed in Table 6.

Using the curve fitting app in MATLAB, we selected time (t) and glucose content (S) as independent and dependent variables, respectively. Custom equation 8 was used for fitting. After adjustment, the fitting curve (Fig. 4d) was obtained. The fitting results were highly accurate, with an $R^2$ value of 0.9992 and an RMSE of 0.1545, effectively reflecting the actual glucose consumption by the bacteria. The coefficients (95% confidence intervals) were determined as follows: $S_0 = 30.44$ (30.12, 30.76), $a = 0.0477$ (fixed interval), $b = 3.917$ (2.41, 5.424), $c = 0.683$ (fixed interval), $M = 1.858$ (1.362, 2.353), and $N = 0.09254$ (0.01189, 0.1732).

By substituting these values into the formula, we expressed the kinetic model for glucose consumption as follows:

This model provides a robust description of the kinetics of glucose consumption during the batch culture of *L. casei* TCS. The high $R^2$ and low RMSE values indicate the reliability and accuracy of the model in predicting glucose consumption, which is essential for optimizing fermentation processes.

## Adsorption of lactic acid by anion resin

The selectivity of alkalescent anion resins for lactic acid is generally high, making them effective for adsorbing lactic acid in fermentation broths. Amberlite IRA 67, a cross-linked acrylic resin, is more hydrophilic than styrene resins, enhancing its selectivity for most organic acids. Based on the kinetic model of product formation, we conducted experiments using different lactic acid concentrations. As shown in Fig. 4b, a smaller amount of resin was insufficient to adsorb all the lactic acid produced during fermentation and improve the fermentation environment. With an increase in the resin dosage, its apparent adsorption capacity decreased, and adsorption efficiency deteriorated. In actual fermentation processes, the resin might continue to adsorb other substances present in the broth. Considering both adsorption capacity and efficiency, a resin dosage of 50 g/L was selected for subsequent experiments. At this concentration, the adsorption amount (q) was 0.138.

**TABLE 6** Coefficient constraint conditions of the substrate consumption kinetic model

| Coefficient | Initialization value | Lower limit | Upper limit |
| --- | --- | --- | --- |
| $S_0$ | 0.0000 | −Inf | Inf |
| $a$ | 0.0477 | 0.0477 | 0.0477 |
| $b$ | 2.0000 | 2 | Inf |
| $c$ | 0.6830 | 0.6830 | 0.6830 |
| $M$ | 0.9000 | −Inf | Inf |
| $N$ | 0.0200 | −Inf | Inf |

To determine the resin's selectivity for lactic acid and simulate lactic acid adsorption in the fermentation broth, we added lactic acid to the broth, followed by 50 g/L of resin (Table 7). We measured lactic acid and glucose concentrations in the fermentation broth before and after adsorption. As shown in Table 8, the resin completed most of the ion exchange within 0.5 h of the reaction. When 50 g/L of resin was added to the MRS solution containing 0.06 mol/L lactic acid, it fully adsorbed the lactic acid. However, the pH value dropped below the initial level in the presence of excessive resin because of the adsorption of other anions after lactic acid. Conversely, when 50 g/L of resin was added to the optimized medium containing 0.1 mol/L lactic acid, lactic acid was not fully adsorbed. Because resin was added during the logarithmic growth phase of the bacteria, the lactic acid content in the system continued to increase with bacterial growth. The lactic acid content in the system was approximately 0.06 mol/L. Thus, we found that the addition of 50 g/L of resin was optimal when the bacteria had been cultured for 8 h.

## Relieving salt stress and improving production efficiency using the ion exchange method and fed-batch fermentation

In this study, we used a combination of ion exchange and fed-batch fermentation methods to enhance the production efficiency of *L. casei* TCS. Initially, *L. casei* TCS was cultured in a batch process without pH control. As illustrated in Fig. 5a, the pH dropped to approximately 5.0 after 8 h of treatment. At this point, the sterilized wet resin was added to the fermenter at a rate of 1.0 BV/h (the optimal resin addition rate is shown in Table 9), and the stirring speed was increased to 100–120 r/min to ensure thorough mixing of the resin with the fermentation broth. This addition led to a significant increase in the pH of the fermentation broth, indicating the effective adsorption of lactic acid by the resin. Simultaneously, the bacteria continued to produce lactic acid. When the pH increased to approximately 5.5, the amount of lactic acid produced exceeded the amount adsorbed by the resin, causing the pH to begin decreasing once again. When the pH value decreased to approximately 5.0, half of the initial amount of resin was added to maintain the pH of the fermentation broth between 5.0 and 5.5.

Throughout the fermentation process, we measured the number of viable bacteria every 2 h. The addition of resin alleviated the effect of acid inhibition, prolonging the logarithmic growth phase of the bacteria by approximately 2 h and thus increasing the number of viable bacteria. As fermentation entered the logarithmic growth phase, a large amount of substrate was consumed. After the addition of the resin, the bacteria sustained logarithmic growth, and the glucose content decreased to less than 15 g/L after 10 h (Fig. 5b). This substrate depletion led to insufficient nutrients to support ongoing metabolic and growth activities, which likely contributed to the observed decline in the number of viable bacteria during the later stages of fermentation.

Fed-batch fermentation was used to further improve production efficiency. As depicted in Fig. 5c, anion resin was added to the fermentation broth at the 8 h mark during batch culture. The number of bacteria in the group treated with the resin was higher than that in the control group not treated with the resin. The addition of the resin effectively prolonged the logarithmic growth phase. When the fermentation continued for 12 h, the number of bacteria further increased, whereas the count decreased in the control group that did not receive additional feeding. Thus, the implementation of fed-batch fermentation considerably enhanced production efficiency. The combined use of fed-batch fermentation and ion exchange alleviated the effects of acid inhibition, salt stress, and substrate limitation, achieving a maximum cell density of 10.01 lg CFU/mL and representing a 9.3-fold increase compared to the basal medium. These results

TABLE 7 Changes in the apparent adsorption capacity of lactic acid with the amount of resin

| Resin usage (g/L) | 0 | 5 | 10 | 20 | 40 | 50 | 60 | 80 |
|---|---|---|---|---|---|---|---|---|
| Adsorption capacity of lactic acid ($C_0-C$) (g/L) | 0 | 0.95 | 1.55 | 3.05 | 5.8 | 6.9 | 8 | 9 |
| Apparent adsorption amount (q) | /[a] | 0.19 | 0.155 | 0.1525 | 0.145 | 0.138 | 0.1333 | 0.1125 |

[a]"/", Not available.

**TABLE 8** Adsorption of lactic acid by resin in the fermentation broth

| Titration acidity of solution (°T) | No lactic acid added (initial value) | 0 h after adding lactic acid | 0.5 h after adding lactic acid | 1 h after adding lactic acid |
|---|---|---|---|---|
| 0.06 mol/L lactic acid added to MRS medium | 47 | 103 | 29 | 21 |
| Medium optimized by adding 0.1 mol/L lactic acid | 41 | 139 | 75 | 65 |

demonstrated that the combined use of ion exchange resin and fed-batch fermentation effectively improved the production efficiency of *L. casei* TCS.

## Conclusion

This study successfully optimized the medium composition and fermentation conditions for the high-density culture of *L. casei* TCS, significantly enhancing production efficiency. By using a combination of single-factor and orthogonal experimental designs and the response surface methodology, we identified the optimal medium components, including glucose, peptone, and sweet corn juice, as well as essential buffer salts and trace elements. Previous studies employing traditional optimization strategies, such as RSM, have reported improvements in LAB culture yields of approximately 4–7-fold (36, 37) under optimized conditions. For example, studies focusing on *Lactobacillus plantarum* (38) and *Lactobacillus rhamnosus* (39) have demonstrated similar enhancements using RSM, but these methods often require extensive experimental runs and are

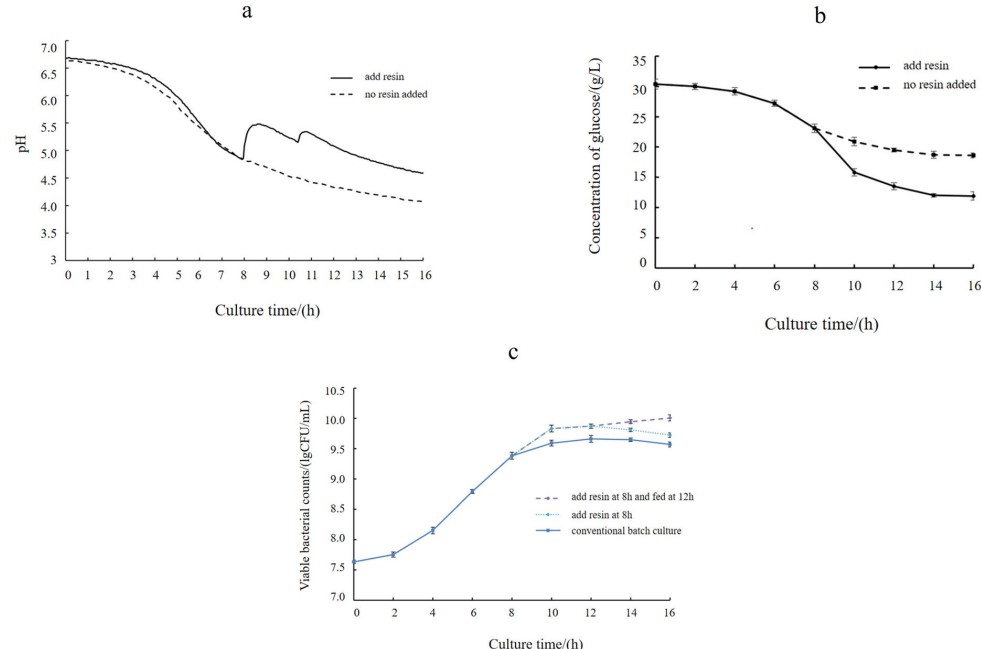

**FIG 5** Changes in pH over time occurring after the addition of resin during fermentation (a). Change in pH value by adding resin culture is shown as the solid line; change in pH value in conventional batch culture is shown as the dashed line. Changes in glucose concentration over time occurring after the addition of resin during fermentation (b). Error bars indicate the standard errors of the mean; the change in glucose contents by adding resin is shown as the solid line; the change in glucose contents in conventional batch culture is shown as the dashed line. Effects of three culture processes on viable bacterial counts of *L. casei* TCS (c). Error bars indicate the standard errors of the mean. Change in viable counts in conventional batch culture is shown as the solid line. Resin was added to the fermentation broth at 8 h, shown as the blue dashed line. Fed-batch fermentation at 12 h is shown as the purple dashed line.

**TABLE 9** The effect of different resin addition rates on the logarithm of viable bacterial counts

| Logarithm of viable bacterial counts (lg CFU/mL) | 8 h | 9 h | 10 h | 11 h | 12 h | 13 h | 14 h | 15 h | 16 h |
|---|---|---|---|---|---|---|---|---|---|
| 0.5B V/h | 9.38 ± 0.03 | 9.55 ± 0.04 | 9.62 ± 0.01 | 9.67 ± 0.03 | 9.74 ± 0.00 | 9.71 ± 0.00 | 9.65 ± 0.04 | 9.62 ± 0.00 | 9.6 ± 0.01 |
| 1.0B V/h | 9.38 ± 0.03 | 9.63 ± 0.00 | 9.76 ± 0.00 | 9.79 ± 0.01 | 9.85 ± 0.03 | 9.76 ± 0.03 | 9.73 ± 0.04 | 9.72 ± 0.04 | 9.7 ± 0.03 |
| 1.5B V/h | 9.38 ± 0.03 | 9.7 ± 0.00 | 9.79 ± 0.01 | 9.79 ± 0.01 | 9.78 ± 0.01 | 9.75 ± 0.04 | 9.71 ± 0.03 | 9.57 ± 0.08 | 9.53 ± 0.03 |

limited in their ability to capture complex, non-linear interactions between variables. The ANN–GA optimization method further refined these conditions, leading to a substantial increase in viable bacterial counts. We effectively modeled the fermentation kinetics of *L. casei* TCS using logistic growth and Luedeking–Piret models. These models provided a comprehensive understanding of cell growth, lactic acid production, and substrate consumption, demonstrating high accuracy and reliability, which are crucial for scaling up fermentation processes. To overcome challenges related to acid inhibition and salt stress, we incorporated Amberlite IRA 67, an anion exchange resin, into the fermentation process. This resin effectively adsorbed lactic acid, maintained optimal pH levels, and extended the logarithmic growth phase of the bacteria, significantly mitigating inhibitory effects and enhancing bacterial growth and cell density. Furthermore, the implementation of fed-batch fermentation in conjunction with the ion exchange resin alleviated the effects of acid inhibition, salt stress, and substrate limitation. This approach resulted in a maximum cell density of 10.01 lg CFU/mL after 14 h of fermentation, representing a 9.3-fold increase compared with the basal medium. Overall, this study provides a robust and cost-effective strategy for the industrial production of *L. casei* TCS. The optimized medium and fermentation processes developed offer significant potential for improving the efficiency and scalability of probiotic production. The optimization approach proposed in this study can be extended to the high-density cultivation of other *Lactobacillus* species, incorporating machine learning algorithms into bioprocess optimization to achieve optimal outcomes. The application of ion-exchange methods for *in situ* lactic acid removal demonstrates superior efficacy compared to traditional chemical neutralization techniques. Furthermore, unlike membrane filtration methods, which require high-end equipment and incur significant maintenance costs, ion-exchange methods are operationally simple and well-suited for industrial-scale production (40). From an economic standpoint, a comprehensive assessment of the cost-effectiveness of resin procurement, regeneration, and reuse is essential, particularly in relation to the potential for reducing operational expenses through enhanced process efficiency and increased product yield (41). Key technical challenges encompass the optimization of resin loading and unloading cycles, the maintenance of consistent ion-exchange capacity, and the assurance of process stability under industrial-scale fermentation conditions (42). Furthermore, the successful integration of this system into established fermentation workflows necessitates seamless compatibility with upstream and downstream operations, alongside the development of robust and scalable control mechanisms to dynamically regulate resin addition and lactic acid removal (41). Addressing these considerations is critical to evaluating the feasibility and industrial scalability of the resin-fed batch system, ultimately positioning it as a promising strategy for improving fermentation efficiency and product quality in large-scale applications.

## ACKNOWLEDGMENTS

This work was supported by the National Natural Science Foundation of China (No. 32402065), the Shanghai Municipal Science and Technology Commission (No. 24ZR1465700), the Science and Technology Commission of Shanghai Municipality (No.

22XD1432700), and the Shanghai Engineering Technology Research Center of Shanghai Science and Technology Commission (No. 20DZ2255600).

The authors declare that they have no known competing financial interests or personal relationships that may have influenced the work reported in this paper.

## AUTHOR AFFILIATION

[1]School of Perfume and Aroma Technology, Shanghai Institute of Technology, Shanghai, Shanghai, China

## AUTHOR ORCIDs

Tianyu Guo ⓘ http://orcid.org/0009-0007-0309-0514
Huaixiang Tian ⓘ https://orcid.org/0000-0002-6097-809X
Chang Ge ⓘ http://orcid.org/0009-0000-5858-5644

## AUTHOR CONTRIBUTIONS

Chen Chen, Conceptualization, Methodology | Tianyu Guo, Methodology, Writing – review and editing | Di Wu, Methodology, Writing – review and editing | Jingyan Shu, Data curation, Methodology | Ningwei Huang, Methodology | Huaixiang Tian, Methodology | Haiyan Yu, Methodology | Chang Ge, Writing – original draft

## DATA AVAILABILITY

No data were used for the research described in the article.

## ADDITIONAL FILES

The following material is available online.

### Open Peer Review

**PEER REVIEW HISTORY (review-history.pdf).** An accounting of the reviewer comments and feedback.

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
