## [Reviewer comments · Microbiology Spectrum]

Microbiology Spectrum

Study on *Lacticaseibacillus casei* TCS fermentation kinetic models and high-density culture strategy

Chen Chen, Tian Guo, Di Wu, Ning Huang, Jing Shu, Huaixiang Tian, Haiyan Yu, and Chang Ge

Corresponding Author(s): Chang Ge, Shanghai Institute of Technology

Review Timeline:

Submission Date:	October 23, 2024
Editorial Decision:	January 22, 2025
Revision Received:	March 4, 2025
Accepted:	March 17, 2025

Editor: Xia Ding

Reviewer(s): Disclosure of reviewer identity is with reference to reviewer comments included in decision letter(s). The following individuals involved in review of your submission have agreed to reveal their identity: Rajib Saha (Reviewer #2); zhihong sun (Reviewer #3)

Transaction Report:

DOI: <https://doi.org/10.1128/spectrum.02590-24>

Re: Spectrum02590-24 (Study on Lacticaseibacillus casei TCS fermentation kinetic models and high-density culture strategy)

Dear Dr. Chang Ge:

Thank you for the privilege of reviewing your work. Below you will find my comments, instructions from the Spectrum editorial office, and the reviewer comments.

Revision Guidelines

Sincerely,
Xia Ding
Editor
Microbiology Spectrum

Reviewer #2 (Comments for the Author):

Summary: This study investigates an innovative optimization framework for the fermentation of Lacticaseibacillus casei TCS. By integrating artificial neural networks (ANN) and genetic algorithms (GA), the authors aim to optimize medium composition and fermentation conditions, achieving significant improvements in bacterial viability and cell density. Additionally, the study incorporates a resin-fed batch process to mitigate lactic acid inhibition. The approach presents potential applications for

enhanced production in probiotic and biotechnological industries.

Major Critiques:

- 1) Introduction, Page 3-5: The introduction explained the importance of optimizing fermentation conditions for lactic acid bacteria (LAB), but it does not provide sufficient justification for employing the ANN-GA framework or integrating the resin-fed system in this specific study. The manuscript does not clearly outline the limitations of traditional optimization approaches (e.g., response surface methodology) or why ANN-GA offers superior advantages in this context. Similarly, the novelty of using a resin-fed system to address lactic acid inhibition is not well-articulated, leaving the rationale for this experimental design unclear. Without these details, the study's contributions appear incremental rather than innovative.
- 2) Results and discussion, Page 20-21: The authors provided some information about the ANN architecture (e.g., a 3-3-1 topology and the Bayesian regularization algorithm); however, critical details are missing. These include the specific activation functions used in the hidden layers and output layer, the method for hyperparameter tuning, the rationale for selecting the architecture, and details on the validation process. The dataset splits (70% training, 15% validation, 15% testing) are mentioned but not linked to performance metrics, and no justification for this split is provided. Additionally, overfitting prevention techniques are not discussed.
- 3) Materials and method, Page 9-10: While the manuscript provides basic GA parameters (population size = 50, crossover probability = 0.8, maximum iterations = 300), it lacks details on how these parameters were optimized or selected. The stopping criteria are vaguely described, and there is no discussion of how the GA was evaluated for convergence. Additionally, the resin-fed system description omits critical operational details such as resin type, ion-exchange capacity, flow rate, resin replacement frequency, and its direct impact on bacterial growth and lactic acid removal. This lack of detail makes the experimental setup difficult to replicate.
- 4) Results and discussion, Page 27-28: The manuscript reports a 9.3-fold increase in bacterial viability, but this finding is not contextualized within the broader field of LAB optimization. The authors do not compare this improvement to similar studies, making it difficult to evaluate its significance. While the general benefits of ANN-GA are noted, the discussion lacks a detailed exploration of how this approach compares to traditional optimization strategies (e.g., response surface methodology) in terms of efficiency, scalability, and reproducibility. Without these comparisons, the broader relevance and novelty of the findings remain unclear.
- 5) Overall: While the manuscript reports RMSE values for ANN training and kinetic model fitting, it does not include additional validation metrics such as MAE or R^2 , which are critical for assessing the reliability of the model's predictions. The absence of cross-validation or testing on an independent dataset raises concerns about the model's robustness and generalizability. Without these validation steps, it is unclear whether the ANN-GA framework can be applied reliably to similar datasets or different experimental conditions.
- 6) Results and discussion, Page 17-19: While the study demonstrates improved bacterial viability, it does not discuss the feasibility of scaling the resin-fed batch system for industrial applications. Important considerations such as cost-effectiveness, technical challenges, and the practical integration of this approach into large-scale fermentation processes are omitted. This omission leaves readers uncertain about the real-world impact and utility of the findings.
- 7) Results and discussion, Page 19: The manuscript employs ANOVA and Tukey's test to analyze the factors influencing viable bacterial counts, reporting that ammonium citrate had the most significant effect ($p < 0.05$). While these methods are appropriate for the study, critical details about the statistical analysis are missing. The manuscript does not provide confidence intervals to quantify the precision of group comparisons, nor does it report effect sizes to evaluate the practical significance of observed differences. Furthermore, while p-values are mentioned, exact values are not reported, and there is no discussion of whether the assumptions of ANOVA, such as normality and homoscedasticity, were checked or met. These omissions reduce the transparency and robustness of the statistical findings.

Minor Critiques:

- 1) Materials and method, Page 12-14: The derivation of kinetic models is presented in a mathematically intensive manner that may alienate readers without advanced expertise in modeling. While the equations are comprehensive, they lack intuitive explanations or graphical representations that illustrate their practical implications. Simplifying the description and including diagrams would make the models more accessible to a broader audience.

Issue of Grammar, Spelling, Proofreading, etc. [note: don't focus on this section, we aren't proofreaders, but if I note something flagrant like misspellings, difficult sentence structure, etc. I'll call it out here]

- 1) Figure Captions, Page 33, Lines 709-711: The manuscript contains typographical errors (e.g., "meaning squared error" in Fig. 3b) and awkward phrasing, affecting readability.

Reviewer #3 (Comments for the Author):

This manuscript investigates the optimization of *Lactocaseibacillus casei* TCS production through medium composition and fermentation strategies, utilizing advanced techniques like response surface methodology and artificial neural networks combined with genetic algorithms. The results show significant improvements in cell density, highlighting potential applications in probiotic production. While the study provides valuable insights, the manuscript would benefit from further refinement to enhance

clarity and overall presentation. With these revisions, the paper could make a meaningful contribution to the field.

1. In the Abstract, the phrase "Amberlite IRA 67, an anion exchange resin, which effectively adsorbed lactic acid and maintained pH levels" uses a non-restrictive relative clause starting with "which," but it lacks a verb, which results in a comma splice. This could be revised to "Amberlite IRA 67, an anion exchange resin, effectively adsorbed lactic acid and maintained pH levels."
2. In the Abstract, the phrase "resulting in a maximum cell density of 10.01 lgCFU/mL, a 9.3-fold increase compared with the basal medium" could be made more concise. The term "lgCFU/mL" should be corrected to "log CFU/mL," which is the standard notation. This issue appears multiple times in the text. Additionally, rephrasing this sentence to "resulting in a maximum cell density of 10.01 log CFU/mL, a 9.3-fold increase over the basal medium" would improve readability and precision.
3. In the Importance section, the term "Lactacaseilus casei TCS" should be corrected to "Lactacaseibacillus casei TCS" for consistency and correctness.
4. In the Introduction section, the sentence "The goal of cultivating Lactobacillus casei TCS at high density is to increase production yields, r..." is incomplete. It needs to be finished to ensure clarity. It is suggested to replace by "The goal of cultivating Lactacaseibacillus casei TCS at high density is to increase production yields, overcome challenges related to acid inhibition, and optimize fermentation processes."
5. In the Materials and Methods section, provide more detailed explanations of the experimental approaches, particularly the artificial neural network-genetic algorithm optimization method. This explanation would help readers unfamiliar with these techniques understand them more thoroughly.
6. In the Results section, the sentence "The combined use of fed-batch fermentation and ion exchange alleviated the effects of acid inhibition, salt stress, and substrate limitation, resulting in a maximum cell density of 10.01 lgCFU/mL, a 9.3-fold increase compared with the basal medium" could be made clearer and more concise. Suggestion: "The combined use of fed-batch fermentation and ion exchange alleviated the effects of acid inhibition, salt stress, and substrate limitation, achieving a maximum cell density of 10.01 log CFU/mL-representing a 9.3-fold increase compared to the basal medium."
7. In the Conclusion section, the sentence "This study presents a robust and cost-effective strategy for the industrial production of L. casei TCS, with significant potential for optimizing probiotic production processes." could be more engaging if it were written in active voice.
8. Throughout the manuscript, any claims regarding the use of optimization methods or biological results should be properly cited. For example, when artificial neural networks or genetic algorithms are mentioned, prior works discussing these methods in the context of microbiology or fermentation should be cited. Providing these references is crucial to ensure scientific rigor and credibility, making the manuscript more robust and well-supported.
9. In the Conclusion, including suggestions for future research directions would benefit the manuscript. For example, consider mentioning scaling up the proposed methods for industrial use.

Response to Reviewers' Comments for Spectrum02590-24R1 /Study on *Lacticaseibacillus casei* TCS fermentation kinetic models and high-density culture strategy

Dear Dr. Xia Ding,

We sincerely appreciate the time and effort that you and the reviewers have invested in evaluating our manuscript, "Study on *Lacticaseibacillus casei* TCS fermentation kinetic models and high-density culture strategy" (Spectrum02590-24R1). We are grateful for the constructive feedback, which has significantly contributed to improving the quality of our work. Below, we provide a detailed response to each comment and describe the corresponding revisions in the manuscript. All changes have been highlighted in the revised version for easy reference.

Reviewer #2

Summary: This study investigates an innovative optimization framework for the fermentation of *Lacticaseibacillus casei* TCS. By integrating artificial neural networks (ANN) and genetic algorithms (GA), the authors aim to optimize medium composition and fermentation conditions, achieving significant improvements in bacterial viability and cell density. Additionally, the study incorporates a resin-fed batch process to mitigate lactic acid inhibition. The approach presents potential applications for enhanced production in probiotic and biotechnological industries.

Major Critiques:

1) Introduction, Page 3-5: The introduction explained the importance of optimizing fermentation conditions for lactic acid bacteria (LAB), but it does not provide sufficient justification for employing the ANN-GA framework or integrating the resin-fed system in this specific study. The manuscript does not clearly outline the limitations of traditional optimization approaches (e.g., response surface methodology) or why ANN-GA offers superior advantages in this context. Similarly, the novelty of using a resin-fed system to address lactic acid inhibition is not well-articulated, leaving the rationale for this experimental design unclear. Without these details, the study's contributions appear incremental rather than innovative.

2) Results and discussion, Page 20-21: The authors provided some information about the ANN architecture (e.g., a 3-3-1 topology and the Bayesian regularization algorithm); however, critical details are missing. These include the specific activation functions used in the hidden layers and output layer, the method for hyperparameter tuning, the rationale for selecting the architecture, and details on the validation process. The dataset splits (70% training, 15% validation, 15% testing) are mentioned but not linked to performance metrics, and no justification for this split is provided. Additionally, overfitting prevention techniques are not discussed.

3) Materials and method, Page 9-10: While the manuscript provides basic GA parameters (population size = 50, crossover probability = 0.8, maximum iterations = 300), it lacks details on how these parameters were optimized or selected. The stopping criteria are vaguely described, and there is no discussion of how the GA was evaluated for convergence. Additionally, the resin-fed system description omits critical operational details such as resin type, ion-exchange capacity, flow rate, resin replacement frequency, and its direct impact on bacterial growth and lactic acid removal. This lack of detail makes the experimental setup difficult to replicate.

4) Results and discussion, Page 27-28: The manuscript reports a 9.3-fold increase in bacterial viability, but this finding is not contextualized within the broader field of LAB optimization. The authors do not compare this improvement to similar studies, making it difficult to evaluate its significance. While the general benefits of ANN-GA are noted, the discussion lacks a detailed

exploration of how this approach compares to traditional optimization strategies (e.g., response surface methodology) in terms of efficiency, scalability, and reproducibility. Without these comparisons, the broader relevance and novelty of the findings remain unclear.

5) Overall: While the manuscript reports RMSE values for ANN training and kinetic model fitting, it does not include additional validation metrics such as MAE or R^2 , which are critical for assessing the reliability of the model's predictions. The absence of cross-validation or testing on an independent dataset raises concerns about the model's robustness and generalizability. Without these validation steps, it is unclear whether the ANN-GA framework can be applied reliably to similar datasets or different experimental conditions.

6) Results and discussion, Page 17-19: While the study demonstrates improved bacterial viability, it does not discuss the feasibility of scaling the resin-fed batch system for industrial applications. Important considerations such as cost-effectiveness, technical challenges, and the practical integration of this approach into large-scale fermentation processes are omitted. This omission leaves readers uncertain about the real-world impact and utility of the findings.

7) Results and discussion, Page 19: The manuscript employs ANOVA and Tukey's test to analyze the factors influencing viable bacterial counts, reporting that ammonium citrate had the most significant effect ($p < 0.05$). While these methods are appropriate for the study, critical details about the statistical analysis are missing. The manuscript does not provide confidence intervals to quantify the precision of group comparisons, nor does it report effect sizes to evaluate the practical significance of observed differences. Furthermore, while p-values are mentioned, exact values are not reported, and there is no discussion of whether the assumptions of ANOVA, such as normality and homoscedasticity, were checked or met. These omissions reduce the transparency and robustness of the statistical findings.

Minor Critiques:

1) Materials and method, Page 12-14: The derivation of kinetic models is presented in a mathematically intensive manner that may alienate readers without advanced expertise in modeling. While the equations are comprehensive, they lack intuitive explanations or graphical representations that illustrate their practical implications. Simplifying the description and including diagrams would make the models more accessible to a broader audience.

Issue of Grammar, Spelling, Proofreading, etc. [note: don't focus on this section, we aren't proofreaders, but if I note something flagrant like misspellings, difficult sentence structure, etc. I'll call it out here]

1) Figure Captions, Page 33, Lines 709-711: The manuscript contains typographical errors (e.g., "meaning squared error" in Fig. 3b) and awkward phrasing, affecting readability.

Response: We sincerely appreciate the time and effort you and the reviewers have invested in evaluating our manuscript, "Study on *Lactocaseibacillus casei* TCS fermentation kinetic models and high-density culture strategy." We have carefully considered all the comments and revised our manuscript accordingly. Below, we provide a point-by-point response to the reviewers' suggestions.

We believe that these revisions have significantly improved our manuscript, and we sincerely appreciate the constructive feedback provided by the reviewers. We hope that the revised manuscript now meets the standards of *Microbiology Spectrum*.

Comment 2:1) Introduction, Page 3-5: The introduction explained the importance of optimizing

fermentation conditions for lactic acid bacteria (LAB), but it does not provide sufficient justification for employing the ANN-GA framework or integrating the resin-fed system in this specific study. The manuscript does not clearly outline the limitations of traditional optimization approaches (e.g., response surface methodology) or why ANN-GA offers superior advantages in this context. Similarly, the novelty of using a resin-fed system to address lactic acid inhibition is not well-articulated, leaving the rationale for this experimental design unclear. Without these details, the study's contributions appear incremental rather than innovative.

Response: We appreciate this valuable comment. In the introduction, I address the limitations of conventional optimization methods, highlighting the significant advantages of the ANN-GA approach in this context. Additionally, I emphasize the innovation of employing a resin feed system to mitigate lactate inhibition. The specific modifications are as follows: For instance, the effectiveness and limitations of the Response Surface Methodology (RSM) in the simulation of discrete and stochastic manufacturing systems have been evaluated. A main disadvantage of RSM is its inability to consider interactive effects among variables, which is crucial for accurately determining output-input relationships. Meanwhile, ANN-GA is highly effective for solving complex, nonlinear, and high-dimensional problems. ANNs model intricate input-output relationships without explicit formulas, while GA explores vast search spaces to identify global optima, overcoming local optima issues inherent in traditional methods. The combination of data-driven learning from ANN and evolutionary optimization from GA provides robustness and flexibility, adapting to various domains without assumptions on data distribution. Additionally, ANN-GA's parallelizability enhances scalability, making it suitable for large-scale or real-time optimization tasks, offering superior performance in dynamic and uncertain environments. The updated content is now included on page [4-6], line [75-87, 93-103].

Comment 2:2) 2) Results and discussion, Page 20-21: The authors provided some information about the ANN architecture (e.g., a 3-3-1 topology and the Bayesian regularization algorithm); however, critical details are missing. These include the specific activation functions used in the hidden layers and output layer, the method for hyperparameter tuning, the rationale for selecting the architecture, and details on the validation process. The dataset splits (70% training, 15% validation, 15% testing) are mentioned but not linked to performance metrics, and no justification for this split is provided. Additionally, overfitting prevention techniques are not discussed.

Response: Thank you for your insightful suggestion. I wish to address your detailed inquiries regarding the ANN architecture as follows, in the Neural Network Toolbox of MATLAB R2018a, an artificial neural network (ANN) model was constructed and trained. The model was designed with a 3-3-1 architecture, comprising an input layer with three neurons, a hidden layer with three neurons, and an output layer with a single neuron. The hidden layer employed the tansig activation function, while the output layer utilized the purelin activation function. During the training process, various training algorithms and different numbers of hidden layer neurons were tested to optimize model performance. Ultimately, the Bayesian Regularization training algorithm was selected, and the number of hidden layer neurons was set to three, yielding satisfactory training results. Additionally, hyperparameters such as the learning rate and the maximum number of training iterations were fine-tuned through experiments to prevent overfitting and enhance the generalization capability of the model. The hyperparameter tuning method employed in this study

was Bayesian Regularization (BR), which automatically adjusts the regularization parameters to balance model complexity and generalization. Additionally, early stopping was implemented by monitoring the validation set error. Training was terminated if the validation error did not decrease for 10 consecutive iterations, preventing overfitting while ensuring optimal model performance. The final ANN architecture (3-3-1) achieved a training RMSE of 5.9327×10^{-5} at the 86th iteration, while the model's R^2 reached 0.9656, demonstrating a strong fitting capability and high predictive accuracy. To ensure the stability of the model, a 5-fold ($k = 5$) cross-validation was employed for model evaluation in this study. The original dataset was randomly divided into five equally sized subsets. In each iteration, one subset was used as the validation set, while the remaining four subsets were used for training. This process was repeated for five rounds. After each training iteration, the root mean square error (RMSE) and the coefficient of determination (R^2) were computed and recorded for the validation set. The final model performance was assessed based on the average values of RMSE and R^2 across the five folds. The experimental results indicated that the ANN model achieved an average RMSE of 6.21×10^{-5} and an average R^2 of 0.9621 across the five-fold cross-validation, with a relatively small standard deviation, demonstrating strong generalization capability across different subsets (see Fig. 4). Furthermore, no single fold exhibited an excessively high RMSE, further confirming that the dataset partitioning was appropriate and that the model did not suffer from severe overfitting or underfitting. The dataset was divided into 70% for training, 15% for validation, and 15% for testing. This split was chosen to ensure a sufficient amount of data for training while maintaining enough validation and test data to assess model performance and generalization. The validation set was used to tune hyperparameters and prevent overfitting, while the test set provided an unbiased evaluation of the final model. We compared the modeling effects of different training algorithms to select the most appropriate one. The training continued until the ANN model met the predetermined training stop criteria, ensuring optimal performance without unnecessary complexity. The fitting and prediction accuracies of the model were evaluated using R^2 and root mean square error (RMSE), linking dataset partitioning to performance assessment. The updated content is now included on page [22-23], line [427-446].

Comment 2: 3) Materials and method, Page 9-10: While the manuscript provides basic GA parameters (population size = 50, crossover probability = 0.8, maximum iterations = 300), it lacks details on how these parameters were optimized or selected. The stopping criteria are vaguely described, and there is no discussion of how the GA was evaluated for convergence. Additionally, the resin-fed system description omits critical operational details such as resin type, ion-exchange capacity, flow rate, resin replacement frequency, and its direct impact on bacterial growth and lactic acid removal. This lack of detail makes the experimental setup difficult to replicate.

Response: We sincerely appreciate the valuable suggestions provided by the reviewer. In response to your comprehensive guidance on optimizing or selecting these parameters, we have provided a thorough supplementation. The updated content is now included on page [11], line [199-220]. In addition, in the part of resin feeding system, additional parameters, including filler type and resin flow rate, have been incorporated. The updated content is now included on page [17], line [328-330] and page [28], line [557-562]. We sincerely apologize for not conducting the resin replacement experiment due to the short fermentation duration. We recognize its significance and

plan to explore it in future research. We extend our sincerest apologies once again for any shortcomings in our work.

Comment 2: 4) Results and discussion, Page 27-28: The manuscript reports a 9.3-fold increase in bacterial viability, but this finding is not contextualized within the broader field of LAB optimization. The authors do not compare this improvement to similar studies, making it difficult to evaluate its significance. While the general benefits of ANN-GA are noted, the discussion lacks a detailed exploration of how this approach compares to traditional optimization strategies (e.g., response surface methodology) in terms of efficiency, scalability, and reproducibility. Without these comparisons, the broader relevance and novelty of the findings remain unclear.

Response: We sincerely appreciate the valuable suggestions provided by the reviewer. Regarding the two issues of this finding not being incorporated into the broader field of LAB optimization and the lack of comparison between this approach and traditional optimization strategies, we have compared our research findings with the current high-density cultivation of lactic acid bacteria and further elaborated on the advantages of the ANN-GA approach. The updated content is now included on page [30-31], line [595-601].

Comment 2:5) Overall: While the manuscript reports RMSE values for ANN training and kinetic model fitting, it does not include additional validation metrics such as MAE or R^2 , which are critical for assessing the reliability of the model's predictions. The absence of cross-validation or testing on an independent dataset raises concerns about the model's robustness and generalizability. Without these validation steps, it is unclear whether the ANN-GA framework can be applied reliably to similar datasets or different experimental conditions.

Response: We sincerely appreciate the valuable suggestions provided by the reviewer. To evaluate the reliability of model predictions, we add the mean absolute error (MAE). MAE (Mean Absolute Error): 0.01. The updated content is now included on page [22], line [427-435]. To ensure the stability of the model, a 5-fold ($k = 5$) cross-validation was employed for model evaluation in this study. The updated content is now included on page [22-23], line [435-446].

Comment 2:6) Results and discussion, Page 17-19: While the study demonstrates improved bacterial viability, it does not discuss the feasibility of scaling the resin-fed batch system for industrial applications. Important considerations such as cost-effectiveness, technical challenges, and the practical integration of this approach into large-scale fermentation processes are omitted. This omission leaves readers uncertain about the real-world impact and utility of the findings.

Response: We sincerely appreciate your valuable suggestions. We have supplemented our discussion on the feasibility of extending the resin-fed batch system to industrial applications, including cost-effectiveness, technical challenges, and the practical integration of this approach into large-scale fermentation processes. The updated content is now included on page [31-32], line [615-638].

Comment 2:7) Results and discussion, Page 19: The manuscript employs ANOVA and Tukey's test to analyze the factors influencing viable bacterial counts, reporting that ammonium citrate had the most significant effect ($p < 0.05$). While these methods are appropriate for the study, critical details about the statistical analysis are missing. The manuscript does not provide confidence

intervals to quantify the precision of group comparisons, nor does it report effect sizes to evaluate the practical significance of observed differences. Furthermore, while p-values are mentioned, exact values are not reported, and there is no discussion of whether the assumptions of ANOVA, such as normality and homoscedasticity, were checked or met. These omissions reduce the transparency and robustness of the statistical findings.

Response: We sincerely appreciate your valuable suggestions. To ensure the applicability of ANOVA, we have incorporated hypothesis testing using the Shapiro-Wilk and Levene's tests. Specific p-values are reported to enhance data transparency, while the effect size (η^2) is included to quantify the magnitude of different factors' impacts. Additionally, a 95% confidence interval (CI) has been provided to improve the robustness of the statistical analysis. The results of Tukey's post-hoc test are explicitly presented to further enhance the clarity and interpretability of the findings. The updated content is now included on page [20-21], line [395-410].

Minor Critiques:

1) Materials and method, Page 12-14: The derivation of kinetic models is presented in a mathematically intensive manner that may alienate readers without advanced expertise in modeling. While the equations are comprehensive, they lack intuitive explanations or graphical representations that illustrate their practical implications. Simplifying the description and including diagrams would make the models more accessible to a broader audience.

Issue of Grammar, Spelling, Proofreading, etc. [note: don't focus on this section, we aren't proofreaders, but if I note something flagrant like misspellings, difficult sentence structure, etc. I'll call it out here]

1) Figure Captions, Page 33, Lines 709-711: The manuscript contains typographical errors (e.g., "meaning squared error" in Fig. 3b) and awkward phrasing, affecting readability.

Response: We sincerely appreciate your valuable suggestions. We have provided corresponding explanations and simplifications of the equations and figures to facilitate a better understanding of the article for the readers. We have provided a more detailed explanation of the relevant figures and thoroughly reviewed the grammar and structure.

Reviewer #3 (Comments for the Author):

This manuscript investigates the optimization of *Lacticaseibacillus casei* TCS production through medium composition and fermentation strategies, utilizing advanced techniques like response surface methodology and artificial neural networks combined with genetic algorithms. The results show significant improvements in cell density, highlighting potential applications in probiotic production. While the study provides valuable insights, the manuscript would benefit from further refinement to enhance clarity and overall presentation. With these revisions, the paper could make a meaningful contribution to the field.

1. In the Abstract, the phrase "Amberlite IRA 67, an anion exchange resin, which effectively adsorbed lactic acid and maintained pH levels" uses a non-restrictive relative clause starting with "which," but it lacks a verb, which results in a comma splice. This could be revised to "Amberlite IRA 67, an anion exchange resin, effectively adsorbed lactic acid and maintained pH levels."

2. In the Abstract, the phrase "resulting in a maximum cell density of 10.01 lgCFU/mL, a 9.3-fold increase compared with the basal medium" could be made more concise. The term "lgCFU/mL" should be corrected to "log CFU/mL," which is the standard notation. This issue appears multiple times in the text. Additionally, rephrasing this sentence to "resulting in a maximum cell density of 10.01 log CFU/mL, a 9.3-fold increase over the basal medium" would improve readability and

precision.

3. In the Importance section, the term "*Lactocaseilus casei* TCS" should be corrected to "*Lactocaseibacillus casei* TCS" for consistency and correctness.

4. In the Introduction section, the sentence "The goal of cultivating *Lactobacillus casei* TCS at high density is to increase production yields, r..." is incomplete. It needs to be finished to ensure clarity. It is suggested to replace by "The goal of cultivating *Lactocaseibacillus casei* TCS at high density is to increase production yields, overcome challenges related to acid inhibition, and optimize fermentation processes."

5. In the Materials and Methods section, provide more detailed explanations of the experimental approaches, particularly the artificial neural network-genetic algorithm optimization method. This explanation would help readers unfamiliar with these techniques understand them more thoroughly.

6. In the Results section, the sentence "The combined use of fed-batch fermentation and ion exchange alleviated the effects of acid inhibition, salt stress, and substrate limitation, resulting in a maximum cell density of 10.01 lgCFU/mL, a 9.3-fold increase compared with the basal medium" could be made clearer and more concise. Suggestion: "The combined use of fed-batch fermentation and ion exchange alleviated the effects of acid inhibition, salt stress, and substrate limitation, achieving a maximum cell density of 10.01 log CFU/mL-representing a 9.3-fold increase compared to the basal medium."

7. In the Conclusion section, the sentence "This study presents a robust and cost-effective strategy for the industrial production of *L. casei* TCS, with significant potential for optimizing probiotic production processes." could be more engaging if it were written in active voice.

8. Throughout the manuscript, any claims regarding the use of optimization methods or biological results should be properly cited. For example, when artificial neural networks or genetic algorithms are mentioned, prior works discussing these methods in the context of microbiology or fermentation should be cited. Providing these references is crucial to ensure scientific rigor and credibility, making the manuscript more robust and well-supported.

9. In the Conclusion, including suggestions for future research directions would benefit the manuscript. For example, consider mentioning scaling up the proposed methods for industrial use.

We sincerely appreciate the time and effort you and the reviewers have invested in evaluating our manuscript, "Study on *Lactocaseibacillus casei* TCS fermentation kinetic models and high-density culture strategy." We have carefully considered all the comments and revised our manuscript accordingly. Below, we provide a point-by-point response to the reviewers' suggestions.

We believe that these revisions have significantly improved our manuscript, and we sincerely appreciate the constructive feedback provided by the reviewers. We hope that the revised manuscript now meets the standards of *Microbiology Spectrum*.

Comment 3:1 , 3:2) 1. In the Abstract, the phrase "Amberlite IRA 67, an anion exchange resin, which effectively adsorbed lactic acid and maintained pH levels" uses a non-restrictive relative clause starting with "which," but it lacks a verb, which results in a comma splice. This could be revised to "Amberlite IRA 67, an anion exchange resin, effectively adsorbed lactic acid and maintained pH levels."

2. In the Abstract, the phrase "resulting in a maximum cell density of 10.01 lgCFU/mL, a 9.3-fold increase compared with the basal medium" could be made more concise. The term "lgCFU/mL"

should be corrected to "log CFU/mL," which is the standard notation. This issue appears multiple times in the text. Additionally, rephrasing this sentence to "resulting in a maximum cell density of 10.01 log CFU/mL, a 9.3-fold increase over the basal medium" would improve readability and precision.

Response: First and foremost, we sincerely appreciate your recognition of our research. Regarding the grammatical suggestions you provided, we have made revisions in the abstract section. The updated content is now included on page [1-2], line [16-17] and [19-22].

Comment 3-3) 3. In the Importance section, the term "*Lacticaseilus casei* TCS" should be corrected to "*Lacticaseibacillus casei* TCS" for consistency and correctness.

Response: Regarding the spelling issues you raised, we have made the necessary corrections. We sincerely apologize for the oversight in our work. The updated content is now included on page [2], line [24-25].

Comment 3-4) 4. In the Introduction section, the sentence "The goal of cultivating *Lactobacillus casei* TCS at high density is to increase production yields, r..." is incomplete. It needs to be finished to ensure clarity. It is suggested to replace by "The goal of cultivating *Lacticaseibacillus casei* TCS at high density is to increase production yields, overcome challenges related to acid inhibition, and optimize fermentation processes."

Response: Regarding the issue you raised concerning sentence clarity, we have made the necessary revisions. The updated content is now included on page [2], line [25-27].

Comment 3-5) 5. In the Materials and Methods section, provide more detailed explanations of the experimental approaches, particularly the artificial neural network-genetic algorithm optimization method. This explanation would help readers unfamiliar with these techniques understand them more thoroughly.

Response: Regarding the issue you raised about the lack of detailed explanations for experimental procedures, we have revised the materials and methods accordingly. We sincerely appreciate this valuable suggestion. The updated content is now included on page [10-11], line [199-220].

Comment 3-6) In the Results section, the sentence "The combined use of fed-batch fermentation and ion exchange alleviated the effects of acid inhibition, salt stress, and substrate limitation, resulting in a maximum cell density of 10.01 lgCFU/mL, a 9.3-fold increase compared with the basal medium" could be made clearer and more concise. Suggestion: "The combined use of fed-batch fermentation and ion exchange alleviated the effects of acid inhibition, salt stress, and substrate limitation, achieving a maximum cell density of 10.01 log CFU/mL-representing a 9.3-fold increase compared to the basal medium."

Response: Regarding the suggestions you provided for sentence structure modifications, we have made revisions in the results section. The updated content is now included on page [29], line [583-586].

Comment 3-7) In the Conclusion section, the sentence "This study presents a robust and cost-effective strategy for the industrial production of *L. casei* TCS, with significant potential for optimizing probiotic production processes." could be more engaging if it were written in active

voice.

Response: Regarding the grammatical suggestions you provided, we have made revisions in the conclusion section. The updated content is now included on page [31], line [615-618].

Comment 3-8) Throughout the manuscript, any claims regarding the use of optimization methods or biological results should be properly cited. For example, when artificial neural networks or genetic algorithms are mentioned, prior works discussing these methods in the context of microbiology or fermentation should be cited. Providing these references is crucial to ensure scientific rigor and credibility, making the manuscript more robust and well-supported.

Response: Regarding the issue you raised concerning the appropriate citation of statements related to the use of optimization methods or biological results, we have revised the conclusion accordingly. We have expanded the existing literature by incorporating additional research on optimization methods and their corresponding biological outcomes. The updated content is now included on page [30], line [595-601].

Comment 3-9) In the Conclusion, including suggestions for future research directions would benefit the manuscript. For example, consider mentioning scaling up the proposed methods for industrial use.

Response: Regarding the issue you raised about including recommendations for future research directions in the conclusion, we have revised the conclusion accordingly. The updated content is now included on page [31-32], line [615-638].

Re: Spectrum02590-24R1 (Study on Lacticaseibacillus casei TCS fermentation kinetic models and high-density culture strategy)

Dear Dr. Chang Ge:

Your manuscript has been accepted, and I am forwarding it to the ASM production staff for publication. Your paper will first be checked to make sure all elements meet the technical requirements. ASM staff will contact you if anything needs to be revised before copyediting and production can begin. Otherwise, you will be notified when your proofs are ready to be viewed.

Sincerely,
Xia Ding
Editor
Microbiology Spectrum

Reviewer #2 (Comments for the Author):

The raised comments and concerns were addressed.